# Physical and Psychological Symptomatology, Co-Parenting, and Emotion Socialization in High-Conflict Divorces: A Profile Analysis

**DOI:** 10.3390/ijerph21091156

**Published:** 2024-08-30

**Authors:** Inés Pellón-Elexpuru, Ana Martínez-Pampliega, Susana Cormenzana

**Affiliations:** Faculty of Health Sciences, University of Deusto, Avenida de las Universidades, 24, 48007 Bilbao, Spain; martinez.pampliega@deusto.es (A.M.-P.); susana.cormenzana@deusto.es (S.C.)

**Keywords:** divorce, conflict, symptomatology, co-parenting, emotion socialization, resilience

## Abstract

Although the consequences of divorce and conflict have been extensively studied, most research has focused on children rather than ex-spouses, although variables such as parental health or co-parenting may have an influence on children’s development through processes such as emotion socialization. In addition, the relationship between these variables has never been considered in high-conflict divorces. Therefore, the present study aimed to analyze the impact of physical and psychological symptomatology and co-parenting on the emotion socialization patterns of parents experiencing high-conflict divorces. Furthermore, the moderating role of resilience was considered, as it has been highly studied as a coping mechanism in adverse situations but barely in divorce at the parental level. For this purpose, a Latent Profile Analysis was carried out with Mplus 8.10, using a sample of 239 parents from Family Visitation Centers. Results revealed, on the one hand, that parents with fewer physical and psychological symptoms sowed more emotion socialization behaviors than those with more symptomatology. On the other hand, in situations of high interparental conflict, the role of co-parenting and resilience seems less relevant than that of physical and psychological symptomatology when analyzing parental skills like emotion socialization.

## 1. Introduction

Currently, a large proportion of married or cohabiting couples undergo divorce or separation. More precisely, in 2021, 700,000 divorces occurred in the European Union [1] and 690,000 across 45 states in the USA in the same year [2]. Divorce often involves financial and legal changes, the renegotiation of parental relationships, changes in friendships and social networks, and moving. Considering parents experiencing high-conflict divorces, if not well managed, these changes can generate tension and conflict that may affect their physical and psychological health [3]. Subsequently, this negative environment and parents’ distress can spill over into their co-parental relationship (Refs. [4,5] and their parenting skills [6]).

However, the extent of these consequences can vary substantially because divorce is a multifactorial phenomenon, and thus, its impact depends on various inter- and intrapersonal factors [7]. In this sense, although it has been demonstrated that most adults can recover well over time [3,8], some situations, such as high levels of destructive conflict, may increase vulnerability to negative consequences [9,10,11,12].

Considering European data, between 5 and 25% of divorces occur with relatively high levels of conflict both during and after the breakup [10]. High-conflict divorces involve constant negative interactions and a hostile and insecure emotional environment. Conflicts preceding, during, and after divorce may reflect relationships characterized by manifestations of hostility, inability to cooperate, and ongoing legal struggles [13]. All of the above entails a number of consequences that have been linked to clinical, legal, and social resource use [12]. Although high-conflict divorces represent a relatively small group of divorces, they occupy 90% of the resources of the family courts [3,10] and are often very challenging for the public administration to handle [11,14].

Regarding the clinical aspect, the existence of fierce and persistent conflicts before, during, and after divorce within the couple has been related to higher physical and psychological consequences [9,11]. Parents experiencing high-conflict divorces tend to show increased levels of anxiety and depression, difficulties trusting new relationships, insensitivity and insecurity, and a greater fear of commitment [3,10,12]. Furthermore, people who are engaged in conflictive relationships tend to have more physical symptoms and pain, fatigue, insomnia, and a worsening of overall physical health [15]. Living in an environment of constant disagreements and hostility can cause a high level of stress that has been linked to impairments in the immune, endocrine, and cardiovascular systems [16,17], as well as a high incidence of somatic symptoms [18,19]. Both the conflict and all of these related consequences in parents can impact their ability to be aware of and respond to their children’s needs, thus affecting not only their individual parenting skills [10] but also other aspects indirectly related to children’s well-being, such as co-parenting [20,21].

Co-parenting refers to how parents communicate and come to arrangements related to childcare, for which it is especially important that parents coordinate and support each other, acting as a team [22,23,24]. Therefore, it is not surprising that in cases of high interparental conflict, where parents have great difficulties in their communication, which has been linked to parental stress, an adequate co-parental relationship is difficult to achieve [3,4,5,10]. At the same time, these difficulties in the co-parental subsystem may also affect children in an indirect way, as they may have an impact on other aspects of the family structure and functioning [25]. 

Considering the Family Systems Theory [26], the different subsystems that are part of the family are interrelated, affecting one another [27]. This is also in line with Katz and Gottman’s (1996) [28] spillover effect. According to these authors, constant disagreements, distress, and tension in the marital subsystem will impede the adequate functioning of the co-parental subsystem. In addition, these difficulties in co-parental functioning may undermine parents’ individual parenting skills. Parents with high levels of distress and difficulties in managing conflicts may tend to use more parenting practices based on punishment and negative emotion socialization patterns characterized by the denial or invalidation of their children’s emotions [29,30]. 

Emotion socialization is the ability of socializing agents, in this case parents, to facilitate their children’s emotional development [31,32]. Thus, it is related to the expression and regulation of one’s own emotions and understanding others’ emotions and the corresponding behaviors [33,34]. This process begins in early infancy and continues throughout childhood and even adolescence. Emotion socialization is characterized by the explicit discussion about emotions (e.g., helping children understand or regulate their own and others’ emotions), reactions to children’s emotions (e.g., responding to them with support or rejection), and the way parents express their own emotions (e.g., warmth versus negativity) [30,31,35]. How parents carry out this process and respond to children’s emotions is critical to their development and has been linked to significant social and emotional maladjustment [36] and even some types of psychopathology [6,37]. 

Parents’ intentionality in managing their children’s emotions is very important. Parents can socialize their children’s emotions both intentionally and unintentionally, through voluntary and involuntary behaviors, by showing their children how they express, manage, and resolve their own conflicts. In this way, they act as role models for their children to understand and manage their emotions. In addition to intentionality, parents’ interpretation of their children’s emotions is also very important. Therefore, based on these two aspects—the intentionality and the interpretation of emotions—parents can socialize their children’s emotions through four parenting styles: (1) Emotion coaching, which is understood as the parent’s acceptance of their children’s emotional expressions and the desire to teach them about emotions; this dimension corresponds to active emotion socialization. (2) Parental acceptance of children’s negative emotions. (3) Parental rejection of children’s negative emotions. And (4) parental feelings of uncertainty or ineffectiveness concerning their emotion socialization, which involves the parents’ acceptance of negative emotions but with little guidance in helping the children work through these emotions [38]. 

Parents’ socializing processes can be affected by multiple factors, such as their own emotion regulation, their physical and mental health, or co-parenting [39]. In this regard, a greater presence of parental symptomatology has been shown to worsen parental emotion socialization. On the one hand, parents who present more anxious and depressive symptoms may not be emotionally available and may be less sensitive to their children’s emotional needs, with difficulties in being supportive of their children’s emotional understanding and expression [37]. These parents engage in more non-supportive responses when socializing their children’s emotions [40]. On the other hand, fatigue, insomnia, body aches, illness, and perceptions of poorer physical health have been shown to diminish the ability to think clearly, make decisions, and respond sensitively to children’s demands and needs [41]. 

At the same time, high parental conflict has often been associated with impaired co-parenting [4,5], which may also be linked to greater emotion socialization difficulties. Research has shown that attitudes such as dismissing the other parent’s parenting and conflictual co-parenting behaviors, characterized by competitiveness and criticism, lead to higher stress levels and may impede being responsive and supportive to their children’s needs. Moreover, when parents engage in conflictive interactions, they tend to focus more on disempowering the other parent than on childcare. At the same time, parents with poor co-parenting or conflictive interactions will probably present more negative responses to their children’s emotional reactions [25].

Given all of the above, it is clear that conflict plays a key role in the well-being of divorced people [12,14], as well as in their co-parenting and parenting skills [10]. However, the relationship between parental symptomatology and co-parenting in parents experiencing high-conflict divorces has barely been considered [42], and how this may affect parents’ ability to socialize their children’s emotions even less so. Parents with a higher level of conflict and, therefore, more conflictual co-parenting may be reflecting a high level of distress (i.e., more anxiety, depression, or more physical symptoms), which has not yet been taken into account in the previous literature on high-conflict divorce, but which may be fundamental when intervening with these families [43]. 

Although most research and interventions in high-conflict divorce have focused on the negative aspects or on trying to reduce conflict, given its important and undeniable role, it is worth noting that other personal resources may also warrant attention. Disciplines such as Positive Psychology advocate focusing on positive emotions, positive personality skills, and positive institutions, including the family, to overcome adversity [44]. In this way, focusing on strengths and not only on the negative aspects of divorce can be very useful, especially when there is a high level of destructive conflict charged with negativity and distress. Resilience is one of the most studied resources in adverse situations [45]. Resilience is considered a coping factor that involves maintaining relatively stable and healthy levels of psychological and physical functioning and regaining optimal levels of well-being when facing stressful life events [46]. Resilient individuals have high emotional stability, can adapt flexibly to unexpected events, and even emerge stronger from them [47]. Resilience has also been linked to emotional management, co-parenting, and parenting skills [48]. Even though resilience has been shown to play a key role in stressful situations, it has hardly been studied in high-conflict divorces. In fact, most studies have focused on the children’s resilience [49] rather than on ex-spouses’ [50].

Following all of the above, this research aims to study, through a profile analysis, the relationship between parents’ physical and psychological symptomatology and co-parenting and parental emotion socialization skills in high-conflict divorces. As the sample comprises high-conflict divorced parents, it can be assumed that co-parenting will not be very high, but previous research has been consistent in its positive relevance to some post-divorce outcomes [21,51]. Therefore, it seems relevant to analyze whether co-parenting can make a difference in families and the consequences of divorce in a person-centered analysis rather than a variable-centered approach. This methodology follows the line of previous studies that have tried to assess the impact of divorce from a multidimensional perspective [46,51]. The main goal of this approach is to capture individual differences through a combination of family variables—parental symptomatology and co-parenting—and subsequently create typologies of individuals. By doing so, we can analyze which combination of variables may best explain parents’ ability to socialize their children’s emotions. 

As a second objective, this study also aims to analyze the moderating role of resilience in the relationship between the parents’ profiles and their emotion socialization behaviors. Resilience is an intrinsic personal resource that acts as a protective factor and modulates the impact of adverse or stressful situations. As it has been studied as a protective mechanism in children of high-conflict divorced parents [49], it might be expected to perform the same function in the parents themselves. However, this has not yet been considered. 

Additionally, the role of some control variables, such as the parents’ gender, the type of custody, the frequency of contact with children, the time elapsed since the divorce, and the length of the relationship, will also be analyzed. For clarity, the term divorce will be used interchangeably for divorced and separated parents. 

More specifically, we expect to find clearly differentiated profiles of parents (Hypothesis 1) and a relationship between these profiles and patterns of emotion socialization (Hypothesis 2). In particular, lower symptomatology is expected to be related to higher emotion socialization, whereas a higher incidence of physical and psychological symptoms will negatively affect parents’ emotion socialization. In addition, we expect to find a moderating role of parental resilience in the relationship between parents’ symptomatology, the co-parental relationship, and their emotion socialization patterns (Hypothesis 3). Resilience, as a coping factor in the face of adverse situations, is expected to help parents face the negative consequences of divorce, so the physical and psychological symptomatology is expected to be lower in resilient parents and thus to be related to better emotion socialization patterns. Identifying this interconnection may help researchers and practitioners better understand the different profiles and characteristics of divorced parents in the context of destructive conflict so that specific needs can be better addressed [50].

## 2. Materials and Methods

### 2.1. Participants

The sample comprised 239 parents, of whom 109 were men and 130 were women, recruited at Family Visitation Centers from Spain. These centers aim to ensure that children have contact with both parents in cases of divorce. Therefore, they try to regulate visits between the child and the non-custodial parent when there is a high level of interparental conflict that impedes parents from reaching an agreement on their own [52]. The sociodemographic characteristics of the sample can be seen in Table 1. For the participants’ inclusion, they had to be separated, regardless of marital status (separation, divorce, singleness, or new relationship), with a high level of interparental conflict, assessed through attendance at the above-mentioned centers, and they could not share the same home with the ex-partner. Finally, they should be capable of understanding and answering the questionnaires. The absence of previous psychiatric pathology was also considered as an inclusion criterion. More than half of the participants had middle-school studies (53.8%) and were actively employed (51.7%). The mean number of children per participant was 2 (SD = 0.857), and the type of custody was exclusive in 52.5% of the cases, mainly of the mothers (79.2%). Most participants had no contact with their ex-partner (72%). In particular, this relationship was non-existent for 70% of the mothers and 75.2% of the fathers. Even if, in general, most parents had daily contact with their children (52.5%), in the case of the fathers, it was more frequent for them to have contact every two weeks (30.3%).

### 2.2. Instruments

The variables gender, type of custody, number of children, educational level, employment, marital status, time since divorce, relationship with ex-spouse, and contact with the child were collected through an ad hoc questionnaire, along with other variables describing each participant. 

Physical symptomatology was measured using the Physical Symptoms Inventory (PSI18; Spector & Jex, 1998 [53]; Spanish Version by Rosario et al., 2014 [54]). The PSI-18 questionnaire assesses somatic and physical health symptoms, considering conditions/states of which a person is aware (e.g., “Headache”, “Sleep problems”, or “Dizziness”) across 18 items. For each symptom, respondents were asked to indicate whether they did not have it, had it, or saw a doctor for it in the past 30 days. Thus, each item has three response choices: 1 (No), 2 (Yes, but I didn’t see a doctor), and 3 (Yes, and I saw a doctor). The total score for physical symptomatology is obtained by summing all responses 2 and 3, with a score of 0 to 18 for each symptom. Regarding reliability, the Cronbach alpha obtained was 0.90, and McDonald’s ω was = 0.91. The reliability of the questionnaire was α = 0.87 for fathers and α = 0.91 for mothers. 

Anxiety and depression symptoms were measured using the Hospital Anxiety and Depression Scale (HADS; Zigmond & Snaith, 1983 [55]; Spanish version. Herrero et al., 2003 [56]). The HADS is a 14-item, self-reporting screening scale that contains two scales that measure symptoms of anxiety and depression: The anxiety scale (7 items) mainly measures symptoms of generalized anxiety disorder (e.g., “I get a sort of frightened feeling as if something awful is about to happen”), and the depression scale (7 items) is mostly focused on anhedonia, the main symptom of depression (e.g., “I still enjoy the things I used to enjoy”). All items were rated on a 4-point Likert scale ranging from 0 to 3, which measures the frequency with which the recorded feelings have been perceived over the last week. The HADS is scored by summing the ratings for the 14 items to yield a total mental health score. The maximum score is 21 points, with a score between 0 and 7 considered normal, between 8 and 10 borderline abnormal, and between 11 and 21 abnormal or pathologic. Regarding reliability, the α obtained was 0.91, and the ω was 0.90 for the total scale, and α = 0.89 and ω = 0.90 for anxiety, and α = 0.82 and ω = 0.81 for depression. The reliability of the whole scale was α = 0.91 for fathers and α = 0.90 for mothers.

Co-parenting was measured using the Divorce–Separation Adaptation Questionnaire (CAD-S; Yárnoz-Yaben & Comino, 2010 [57]). This questionnaire consists of 20 items and evaluates the adaptation to divorce of the whole family, completed by one of the parents experiencing high-conflict divorces. The instrument has four dimensions: psychological difficulties of adjustment to divorce, conflict with the ex-partner, negative consequences of separation, and willingness to co-parent. The subscale used in this study was willingness to co-parent (5 items), which describes perceptions of the ex-spouse’s level of cooperation to act for the benefit of their children, helping and supporting the other parent when necessary (e.g., “My ex-partner helps me to raise our children”). All items are rated on a 5-point Likert scale ranging from 1 (strongly disagree) to 5 (strongly agree). There is no total score for the whole questionnaire, but each subscale is considered separately. In the case of co-parenting, the maximum score would be 5, and a score above 2 would indicate difficulties in the functioning of this family subsystem. In this study, the reliability obtained by willingness to co-parent was α = 0.76 and ω = 0.76. The reliability was α = 0.70 for fathers and α = 0.70 for mothers.

Emotion socialization was measured with the Emotion-Related Parenting Styles Self-Test (ERPS; Paterson et al., 2012 [58]), a 20-item short-form questionnaire that produces scores on four different emotion-related parenting styles. Emotion coaching (5 items) assesses a parent’s acceptance of their child’s emotional expression and the desire to teach the child about emotions (e.g., “When my child is sad, I try to help them explore what is making them sad”). Parental rejection of negative emotion (5 items) measures the degree to which parents reject their children’s experience of negative emotions (e.g., “When my child gets angry, my goal is to get them to stop”). Parental acceptance of negative emotion (5 items) assesses the extent to which parents tend to accept negative emotions but provide little guidance in helping the child work through those emotions (e.g., “I think it’s good for kids to feel angry sometimes”). Feelings of uncertainty/ineffectiveness in emotion socialization (5 items) assesses whether parents feel uncertain or ineffective about handling their child’s experience of negative emotions (e.g., “When my child is angry, I’m not quite sure what they want me to do”). All items are rated on a 5-point Likert scale ranging from 1 (always false) to 5 (always true). Total subscale scores are calculated by summing the items of each subscale. Higher scores represent endorsement of the associated parenting style. In this study, the reliability obtained by each subscale was α = 0.97 and ω = 0.97 for emotion coaching; α = 0.97 and ω = 0.97 for parental rejection of negative emotion; α = 0.96 and ω = 0.96 for acceptance of negative emotion; and α = 0.97. and ω = 0.97 for feelings of uncertainty/ineffectiveness. The total reliability score for this questionnaire was α = 0.99 and ω = 0.99. The reliability of the questionnaire was α = 0.75 for fathers and α = 0.80 for mothers.

Resilience was examined using the 14-item Resilience Scale (RS-14; Sánchez-Teruel & Robles-Bello, 2014 [59]). This scale is based on the Resilience Scale (RS-25) by Wagnild and Young (1993) [60]. It measures the degree of individual resilience, which is considered a positive personality characteristic that allows people to adapt to adverse situations. The RS-14 measures two factors: Personal Competence (11 items), which assesses self-confidence, independence, decisiveness, resourcefulness, and perseverance (e.g., “I usually manage in one way or another”), and Self- and Life Acceptance (3 items), which includes adaptability, balance, flexibility, and a stable outlook on life (e.g., “I am a person with an adequate self-esteem”). All items are rated on a Likert scale ranging from 1 (strongly disagree) to 7 (strongly agree). To calculate the scale’s score, the total score of the questionnaire was averaged. The range of scores is between a minimum of 24 and a maximum of 28 points, with a mean score of 71 (SD = 32.81). In this study, the reliability of the Resilience Scale was α = 0.89 and ω = 0.89. The reliability was α = 0.89 in the case of fathers and α = 0.89 for mothers. 

#### 2.2.1. Procedure

This study was carried out during 2022–2023 with the approval of the University’s Ethics Committee (ETK-38/21-22), and it also complied with the original Declaration of Helsinki. It was developed in Spain and included more than 30 national Family Visitation Centers. All existing national centers for which contact information was available were contacted through email and phone calls to inform them about the project. Meetings were arranged for those whose managers showed interest in detailing the aim, requirements, and confidentiality, as well as resolving possible doubts. Necessary agreements were also requested. All the centers had permission and support from the municipalities and related ministerial services. The final participation rate was 60%. The rest of the centers claimed work overload or incompatibility due to participation in other research projects.

After these initial meetings, a detailed information document was given to the corresponding professionals so that they could share it with the families. In these sheets, the main goals of this study, the voluntary nature of their participation, and the confidentiality of their data were detailed. In addition, the technicians were informed of the inclusion criteria so that they could check them in the clinical history and/or the judicial referral report. After the parents agreed to participate, they had to fill out an informed consent form before completing the questionnaires. The questionnaires were completed individually, with an approximate duration of 20 min. The confidentiality of the participants was maintained at all times, with no other identifying information besides the email address or telephone number.

#### 2.2.2. Data Analysis

The present study examines whether parents’ symptomatology (physical and psychological) and co-parenting are related to emotion socialization. Different latent profiles based on psychological and physical symptomatology were developed for this aim through a Latent Profile Analysis (LPA) [61] using Mplus 8.10 [62]. 

In particular, we followed the three-step method proposed by Asparouhov and Muthén (2014) [63]. First, we compared LPA models with one to four latent profiles to estimate the number of underlying profiles. The best model was established considering some indicators such as Akaike’s Information Criterion (AIC), the Bayesian Information Criterion (BIC), the mean-adjusted Bayesian Information Criterion (aBIC), the entropy, the Lo–Mendell–Rubin adjusted likelihood ratio test (LMRa), and the bootstrap likelihood ratio test (BLRT) [64]. To decide the number of profiles, we also considered parsimony, the theoretical interpretation, and the size of the profiles [65]. The second step was to identify the most likely profile for each individual based on the probability of belonging to each profile found in the previous step (Hypothesis 1). In the third step, we analyzed the antecedents and outcomes of the obtained profiles. Regarding the antecedents, we tested whether physical and psychological symptomatology and co-parenting influenced the likelihood of belonging to a certain profile while controlling for parents’ gender, type of custody, frequency of contact with the children, time elapsed since divorce, and duration of the marriage. At the level of outcomes, which is related to Hypothesis 2, we analyzed the predictive differences of each profile in the four dimensions of emotion socialization (emotion coaching, feelings of uncertainty/ineffectiveness in emotion socialization, approval of negative emotions, and rejection of negative emotions). 

After analyzing the profiles, we tested the moderator role of parental resilience to explain the differential levels of emotion socialization as a function of parental anxiety, depression, and physical symptomatology (Hypothesis 3). For this purpose, we tested the interactions between resilience and the classes obtained in the previous LPA analyses.

## 3. Results

First, we calculated the descriptive statistics for the total sample, which is displayed in Table 1. We also used Harman’s single-factor test to test for the common method bias. This may help to ensure the reliability of the research findings, as different self-report questionnaires were used in this study. The results showed that the total variance extracted by one factor was 18%, thus less than 50%, meaning that the common method bias is not present in our study. Second, in the LPA, although the two-profile model showed a significant BLRT and a higher level of entropy compared to the three- and four-profile models, all other indicators were worse. In addition, the level of BLRT was better in the three-profile model, being a preferential value when choosing one model or the other. Finally, even if the four-profile model showed a better fit considering LMRa and aBIC when the distribution of the classes was checked, we found that the fourth class consisted of 0.08% of the sample, so it was not sufficiently representative. Also, the entropy was slightly better in the three-profile model, so it was selected (Table 2).

Subsequently, we observed the profiles’ characteristics (Table 3 and Figure 1) and composition (Table 4). Profile 1 represented 52% of the sample and showed the lowest levels of anxiety, depression, and physical symptomatology and the highest level of co-parenting. Profile 2 represented 36% of the sample and showed the lowest levels of co-parenting. Although these levels of physical and psychological symptomatology of Profile 2 were higher than those of Profile 1, they were lower than those of Profile 3. Finally, Profile 3 represented 12% of the sample and showed the highest level of anxiety, depression, and physical symptoms and also slightly higher levels of co-parenting. In all profiles, the target variables were significant. To conclude, these results indicate the presence of three differential post-divorce profiles, which supports Hypothesis 1.

In the third step of the LPA, we analyzed the antecedents and outcomes of the profiles (see Table 5). Regarding antecedents, using the three-step procedure, the multinomial logistic regression analyses showed that the participants’ gender predicted a higher probability of men belonging to Profile 1 versus Profiles 2 and 3. The rest of the factors did not yield any differences concerning membership in a specific profile.

The main outcome considered was emotion socialization. We measured four types of emotion socialization (emotion coaching, feelings of uncertainty or ineffectiveness in emotion socialization, parental acceptance of negative emotions, and parental rejection of negative emotions). Profile 1 (M = 13.61, SE = 1.13), with fewer symptoms and higher co-parenting, was more likely to employ active emotion socialization patterns or emotion coaching than Profile 2 (M = 6.07, SE = 0.61) and Profile 3 (M = 5.53, SE = 0.48). There were no differences between Profiles 3 and 2 in emotion coaching. In addition, we also found differences in parental rejection of negative emotions, with Profile 1 (M = 11.16, SE = 1.00) again more likely to present rejection of negative emotions than Profile 2 (M = 4.57, SE = 0.47) and Profile 3 (M = 8.19, SE = 7.75). At the same time, Profile 2, which presented more symptomatology than Profile 1 but also the lowest level of co-parenting, was more likely to use negative emotion rejection behaviors than Profile 3. As for the other dimensions of emotion socialization, no significant differences were found between the three profiles.

Finally, based on the differences between the profiles, we tested whether resilience played a moderating role in the relationship between each profile and emotion socialization. Concerning the results obtained from the resilience interactions with each profile, and taking Profile 3 as a reference, neither in rejection of negative emotions (Profile 1: B = 0.07, SE = 0.15; Profile 2: B = −0.03, SE = 0.14) nor in emotion acceptance (Profile 1: B = 0.07, SE = 0.21; Profile 2: B = −0.09, SE = 0.18) were the interactions significant. This was also observed in emotion coaching (Profile 1: B = 0.19, SE = 0.21; Profile 2: B = 0.08, SE = 0.12) and feelings of uncertainty/ineffectiveness in emotion socialization (Profile 1: B = 0.07, SE = 0.20; Profile 2: B = 0.03, SE = 0.20), so the moderation hypothesis could not be confirmed.

## 4. Discussion

The present study makes an important contribution to our understanding of the impact of high-conflict divorce through a person-centered analysis and identifies concurrent patterns in parents’ emotion socialization. To date, this is the first study to consider the relationship between parental symptomatology, co-parenting, and emotion socialization in a sample of high-conflict divorced people.

The results obtained partly support the proposed hypotheses. First, we identified three clearly differentiated profiles based mainly on parental physical and psychological symptomatology (anxiety and depression) (Hypothesis 1). The most frequent was Profile 1 (52%), characterized by low physical and psychological symptomatology. Profile 2 also represented a significant amount of participants (36%) and had mid-levels of parental symptomatology. Finally, Profile 3 comprised the smallest number of participants (12%) and showed the highest levels of anxiety, depression, and physical symptomatology. Moreover, men were more likely to belong to Profile 1. This is not surprising because, as has been reported in the prior literature, women tend to have a higher level of symptomatology after divorce [66,67], especially when children are involved [68].

Regarding the distribution of the profiles (Hypothesis 2), the results indicate a relationship between parental symptomatology and emotion socialization. On the one hand, the profile with the lowest psychological and physical symptoms (Profile 1) was related both to higher emotion coaching and higher rejection of negative emotions toward their children. A priori, this result might seem contradictory, as emotion coaching refers to parents’ acceptance of their children’s emotional expression and the desire to teach them about emotions (e.g., “when they are sad, I try to help them explore what makes them sad”), in contrast to parents’ rejection of children’s experience of negative emotions (e.g., “children acting sad are usually just trying to get adults to feel sorry for them”) [38,58]. 

The previous literature has shown that parents with less symptomatology will probably be more capable of attending to their children’s emotions and actively trying to help them identify, understand, manage, and express them [37]. However, it is important to remember that although Profile 1 showed a lower level of symptomatology and a higher ability to process and accept their own and their children’s emotions [6,30], these parents still present a high level of conflict. Being more conscious of their own emotions while living in a context of constant conflict with the ex-partner, charged with negative emotionality, can make them feel overwhelmed [69,70], leading to a loss of effective parenting practices. In this case, even if the parents have active patterns when socializing their children’s emotions, they may also feel greater rejection when their children express negative emotions (anger and sadness). They may even think children use them to get attention [38].

On the other hand, Profile 2 showed lower levels of symptomatology but also higher rejection of negative emotions than Profile 3. This is in line with the mentioned idea of less symptomatology being linked to more emotion socialization behaviors [30,71], even if they are not always effective. In contrast, parents with higher levels of symptomatology, as in Profile 3, may be so immersed in their own conflicts and distress that they cannot pay attention to their children’s emotional needs [72], respond to their negative emotions [30], or even to make some attribution or seek an explanation for these emotional states. Moreover, it has been found that this attitude is even more common among women [73], which is also in line with the findings of our study.

Regarding co-parenting, the results were contrary to our expectations. Although the level of perceived co-parenting was low in 80% of the cases, there were hardly any differences in the distribution of the participants in the three profiles. Although Profile 3 scored slightly higher than the other two profiles in their co-parenting, this difference was not substantial and did not affect parents’ emotion socialization skills. This distribution is consistent with other previous studies highlighting that when there is a high level of destructive conflict, the need to maintain co-parental contact may lead to an increase of hostility and tension between the ex-partners rather than mitigating it, as they tend to use childcare-related issues as another source of conflict and mutual criticism [20,74]. However, from a Family Systems perspective [26], the co-parental and parental subsystems, even if interrelated, are independent [25]. Thus, even when the parents have interpersonal conflict, if they do not focus on constantly trying to maintain the contact, and their level of stress and subsequent symptomatology is low enough to function adequately, they will probably be more aware of and respond better to their children’s needs [48]. Therefore, the results suggest that when the level of conflict and symptomatology is high, it is probably not the best moment to focus on co-parenting arrangements, and it may be necessary to prioritize intervention on these variables beforehand.

Finally, we considered the moderating role of resilience (Hypothesis 3), but we found no significant result; that is, resilience does not seem to moderate the relationship between any of the three profiles and emotion socialization. The conflictive nature of the sample may partly explain this, as the magnitude of the conflict and subsequent symptomatology may lead to an emotional burnout that outweighs the protective effect of resources such as resilience [69,75]. However, this does not mean that these families are not resilient, as the scores obtained show that they perceived themselves as having a high level of resilience. Instead, we can conclude from all these results that the level of destructive conflict and the subsequent symptomatology is what makes the real difference in parental functioning and emotion socialization [6,30,71]. 

Despite the above, this study is not without limitations, and its results should be taken with caution. First, most of the participants had been divorced for more than three years, and, nevertheless, they are still coming to the Family Visitation Centers. It might be interesting to compare parents at different times after divorce, controlling for the time elapsed, and see how the chronification of the conflict may affect the development of subsequent symptomatology or the co-parental relationship and their resilience. Along the same line, it would be of interest to consider this relationship in divorced parents without conflict, as co-parenting and resilience in this study were not significant, probably due to the level of conflict and symptomatology [20,75]. In addition to the conflict, the socioeconomic profile of the families attending these centers and, more specifically, that of the sample used in this study is characterized by a low level of education, which may affect the generalizability of the results. Therefore, it would be convenient for future studies to include families with more diverse profiles belonging to different contexts so as to enrich the scope of the findings. Second, another main limitation is the cross-sectional nature of this study, which does not allow for the establishment of causal associations. 

Finally, it should be noted that this study only considered parents’ perceptions of their skills in socializing their children’s emotions. Nevertheless, as emotion socialization has a fundamental impact on children’s well-being and emotional development [29], it would be worthwhile for future research to consider their point of view as well.

In spite of these limitations, this study offers important implications for future research and clinical practice. First, it is worth noting that, to our knowledge, this has been the first study to analyze emotion socialization in high-conflict divorces to date. High-conflict divorces are a difficult population to access, so this study makes an important contribution to better understanding the needs and characteristics of these families, which may open the door for future studies to continue deepening this relationship between variables that had never been considered before. Further research is encouraged to analyze this relationship considering other populations, such as parents experiencing divorces without conflict, and some factors, such as gender, which the previous literature has deemed to be controversial [15] but which seems to play a role in parental functioning.

In addition, our findings can help practitioners working with people in high-conflict divorces to better tailor interventions to their specific needs and characteristics, regardless of the context (legal, clinical, or community) [51]. It is important to bear in mind that, in high-conflict divorces, maintaining interparental contact may not be beneficial but instead can actually increase the tension and associated distress [20]. Therefore, it is essential to reduce the destructiveness of the conflict and, especially, the associated parental symptomatology [21] before focusing on establishing any co-parental communication and arrangements.

## 5. Conclusions

In short, it is important to note that when analyzing high-conflict divorce, both research and clinical practice should focus not only on the conflict but also on the parents’ symptomatology and well-being. Parent’s physical and psychological symptoms seem to be key to adjustment to divorce and parenting skills. Keeping this in mind may help health professionals generate preventive and intervention programs that promote an adequate emotional environment so that children can develop emotionally after divorce.

## Figures and Tables

**Figure 1 ijerph-21-01156-f001:**
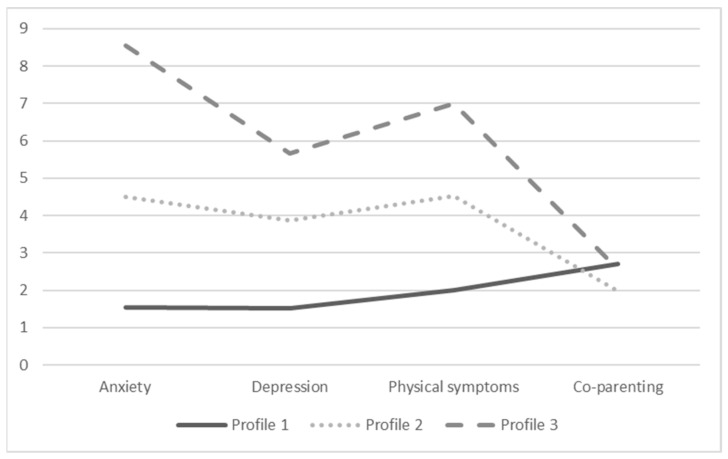
Characteristics of the profiles.

**Table 1 ijerph-21-01156-t001:** Sociodemographic characteristics of the sample.

	Mothers (*n* = 130)		Fathers (*n* = 109)		Total (*n* = 239)	
	M	SD	M	SD	M	SD
Number of children	2	0.93	2	0.76	2	0.62
	n	%	n	%	n	%
Educational level						
Less than primary education	3	2.3	7	6.4	10	4.2
Primary studies	64	49.2	64	58.7	129	53.8
Secondary studies	8	6.2	3	2.8	11	4.6
BUP or FP	20	15.4	16	14.7	36	15
Medium career	10	7.7	6	5.5	16	6.7
Higher career	11	8.5	9	8.3	20	8.3
Master’s or PhD	13	10	4	3.7	17	7.1
Employment						
Unemployed	17	13.1	5	4.6	22	9.2
Active	57	43.8	66	60.6	124	51.7
Retired	9	6.9	3	2.8	12	5
Disabled	17	13.1	14	12.8	31	12.9
Sick leave	10	7.7	9	8.3	19	7.9
Other	18	13.8	9	8.3	27	11.3
Marital status						
Single	57	43.8	46	42.2	103	43.1
Separated	10	7.7	4	3.7	14	5.9
Divorced	49	37.7	38	34.9	87	36.4
Married, not a first marriage	11	8.5	13	11.9	24	10
Common law couple	1	0.8	3	2.8	4	1.7
Time since separation						
Not separated yet	10	7.7	12	11	23	9.6
Only submitted the papers	4	3.1	1	<1	5	2.1
Less than two months	-	-	1	<1	1	<1
Two to six months	3	2.3	3	2.8	6	2.5
Six months to one year	10	7.7	5	4.6	15	6.3
One to two years	13	10	11	10.1	24	10
Two to three years	13	10	16	14.5	29	12.1
More than three years	48	36.9	34	31.2	82	34.2
Relationship with ex-spouse						
Non-existent	91	70	82	75.2	173	72.4
Very scarce	9	6.9	5	4.6	14	5.9
Restricted	21	16.2	16	14.7	37	15.5
Fluid	5	3.8	4	3.7	9	3.8
Type of custody						
Shared	9	6.9	12	11	21	8.8
Exclusive	103	79.2	23	21.1	126	52.5
Non-custodial	18	13.8	73	67	92	38.3
Contact with the child						
Every day	103	79.2	23	21.1	126	52.5
Several days a week	7	5.4	22	20.2	29	12.1
Once a week	6	4.6	22	20.2	28	12.1
Every two weeks	11	8.5	33	30.3	44	18.3
Every month	-	-	1	<1	2	0.8
Less frequently	3	2.3	8	7.3	11	4.6

Note: BUP = Spanish high school; FP = Spanish vocational training.

**Table 2 ijerph-21-01156-t002:** Model fit and model comparisons of Latent Profile Analyses.

Model	AIC	BIC	aBIC	Entropy	LMRa	BLRT
1-profile model	5701.030	5728.517	5703.517	_	_	_
2-profile model	5471.703	5516.952	5475.745	0.865	230.900 ***	239.327 ***
3-profile model	5425.321	5487.972	5430.917	0.827	54.397 ***	56.382 ***
4-profile model	5406.044	5486.098	5413.194	0.800	28.246	29.277 ***

Note: AIC = Akaike’s Information Criterion; BIC = Bayesian Information Criterion; aBIC = adjusted Bayesian Information Criterion; LMRa = Lo–Mendell–Rubin adjusted likelihood ratio test; BLRT = bootstrap likelihood ratio test; _ = no information. *** *p* < 0.001.

**Table 3 ijerph-21-01156-t003:** Descriptive statistics of each profile.

	Anxiety		Depression		Physical		Co-Parenting	
	M	SD	M	SD	M	SD	M	SD
Profile 1	3.24	2.18	3.22	2.67	3.59	3.09	5.40	5.00
Profile 2	9.43	2.28	8.10	3.56	8.15	4.31	3.96	3.97
Profile 3	17.96	1.89	11.88	4.87	12.62	3.48	5.12	5.93

Note: Physical = physical symptomatology.

**Table 4 ijerph-21-01156-t004:** Estimates of the 3-profile model.

	Profile 1		Profile 2		Profile 3	
	Estimate	*p*	Estimate	*p*	Estimate	*p*
Anxiety	1.39	<0.001	3.89	<0.001	7.50	<0.001
Depression	1.00	<0.001	2.40	<0.001	3.61	<0.001
Physical	0.97	<0.001	2.25	<0.001	3.49	<0.001
Co-parenting	1.13	<0.001	0.86	<0.001	1.07	<0.001
Profile n	126		86		36	
% of the sample	53%		36%		12%	

Note: Physical = physical symptomatology.

**Table 5 ijerph-21-01156-t005:** Profile comparison: antecedents and outcomes.

Antecedents	Profile 1 vs. 3		Profile 2 vs. 3		Profile 1 vs. 2	
	β	Adj. ORª	β	Adj. ORª	β	Adj. ORª
Gender	−1.29 ***	0.28 ***	−0.02	0.98	−1.26 ***	0.28 ***
Type of custody	0.42	1.52	0.11	1.11	0.32	1.37
Time since separation	0.24	1.27	0.24	1.27	−0.00	1.00
Time of relationship	−0.03	0.98	0.01	1.01	−0.03	0.97
Frequency of contact	0.29	1.34	0.12	1.13	−0.54	1.19
Outcomes	Profile 1 vs. 3		Profile 2 vs. 3		Profile 1 vs. 3	
	χ^2^	*p*	χ^2^	*p*	χ^2^	*p*
Emotion coaching	34.66	0.000	0.49	0.485	42.95	0.000
Feelings of uncertainty/ineffectiveness	0.32	0.574	0.78	0.378	0.26	0.612
Negative emotion rejection	35.57	0.000	16.77	0.019	5.49	0.000
Negative emotion approval	0.33	0.565	2.55	0.110	0.18	0.673

Note: Adj. ORª = Adjusted Odds Ratio. *** *p* < 0.001. χ^2^ = chi-square.

## Data Availability

Data are contained within the article.

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
