# Peer review of "Physical and Psychological Symptomatology, Co-Parenting, and Emotion Socialization in High-Conflict Divorces: A Profile Analysis"

_ijerph, 2024, doi:10.3390/ijerph21091156_

Round 1

Reviewer 1 Report

Comments and Suggestions for Authors

I appreciate the opportunity to read and review the manuscript, " Physical and psychological symptomatology, co-parenting, and emotion socialization in high-conflict divorces: A profile analysis".

This study had two goals: 1) conduct a latent profile analysis with the combination of three variables (physical health, mental health, and coparenting) among a group of separated parents with a high level of interpersonal conflict; and 2) analyze the moderating role of resilience in the relationship between the profiles and parents’ emotion socialization. The authors were able to identify three profiles, whit different relations to emotional socialization. Resilience did not moderate the relationship between profiles and emotional socialization.

This is an interesting topic and I value the authors´ recommendations for intervention; however, there are several issues that limit enthusiasm for its current form.

My major concern is the lack of a theoretical foundation guiding this study; it is not clear why the authors selected the variables included in the analysis. What is the rationale behind that process?

Only in the discussion de authors refer to a theoretical framework (family systems theory). Perhaps, this theory can be used to frame the study.

Introduction

The introduction is confusing to read. The authors include different concepts, which are not clearly defined and compared (for example, poor coparenting and parental conflict). Please explain how both concepts differ.

Much is presented on parental conflict, parenting skills, but less on the variables used in the analysis (specially on physical symptoms).

I suggest being consistent with the concepts used. For example, it seems the authors use the concepts of parenting practices, parents’ socialization and parental functioning as interchangeable. This might be confusing for the readers, specially when  

In addition, in the introduction the author include statement such as “Between 5 - 25% of divorces occur with relatively high levels of conflict both during 46 and after the breakup (Hald et al., 2019)- line 46-7 ” or “…they occupy 90% of the resources of the family courts 53 (Hald et al., 2019; Ordway et al., 2020) and are often very challenging for the public 54 administration to handle (Haddad et al., 2016; Nikupeteri & Laitinen, 2022). Line 53-55 but they don´t clarify where these data come from. Are the from Spain (where the study was conducted)? If not, are they comparable to Spain?.

Co-parenting is an important variable in the study; however, it is poorly defined. The central aspect of coordination between parents is missing. Moreover, it is surprising that the authors don’t mention any of the work by Feinberg or McHale

Hypothesis

About the hypothesis I suggest being more specific, for example including possible relationships between variables in their directions.

Material and Methods

The authors describe the sample as experiencing high levels of interpersonal conflict. Please explain how this variable was operationalized and assessed. My understanding is that this was a variable not assessed by the authors, but by the visiting program. It would be interesting to see how the distribution of levels of conflict in these families. In the discussion the authors say that the families in this sample present high levels of conflict. What does it mean?

For the instruments, please add reliability for fathers and mothers, and add information whether the instruments have been validated in Spain.

Results

Line 379: the authors report gender differences, but don’t specify whether they refer to male or female.

Please include in the result section the information presented on line 467-8 (that coparenting score did not vary significantly among profiles). Please do the same for scores on resilience. The information that participants scored high in resilience in only provided at the end of the manuscript (in the discussion section).

Discussion

The discussion of the results of profile 1 and use of coaching and rejection is not clear. The author say that these parents are more conscious of their own emotions. What evidence do they use to support this claim?

On line 456-457 the authors say that parents in profile 3 have higher levels of conflict. Higher compared to what?

Mikolajczak and Roskam refer to parental burnout, which is a specific concept. The way the authors (line 461-3) use this concept (naming it emotional burnout) and relate it to their findings is, in my opinion, not accurate.

Please include more discussion on the profiles. Profile 3 is interesting, as it presents high symptoms but also high co- parenting.

Other minor issues

Please revise sentence in line 60. What do the authors mean by “sensitivity”? In parenting research being sensitive is generally characterize as a positive aspect.Please revise table 5b. the title for the first column seems to be incorrect.

Comments on the Quality of English Language

I am not expert in latent profile analysis, so I did not make a detailed review of the methods section. 

Author Response

We appreciate the thorough feedback given by the reviewers. It helped us improving both the content and writing of the manuscript. We would also like to acknowledge the positive words regarding the interest and implications of the study for future research and clinical practice.

 All the adjustments in response to the suggestions of the reviewers are highlighted in yellow in the revised manuscript (complete manuscript resubmission) and this document indicates exactly on which lines they can be found.

Comment 1: My major concern is the lack of a theoretical foundation guiding this study; it is not clear why the authors selected the variables included in the analysis. What is the rationale behind that process?

Only in the discussion de authors refer to a theoretical framework (family systems theory). Perhaps, this theory can be used to frame the study.

Response 1: We appreciate the reviewer's suggestion to use Family Systems Theory as a framework for the study, and have tried to incorporate it into the introduction (lines 74-78).

Regarding the decision of the variables used, we consider that we have carried out an exhaustive analysis of them, relating them to high-conflict divorces and their importance in the patterns of emotion socialization, which is key for the correct emotional development of children (line 57 in advance). Due to limitations in the extension of the article, we were not able to include more information in the introduction part.

 Introduction

Comment 2: The introduction is confusing to read. The authors include different concepts, which are not clearly defined and compared (for example, poor coparenting and parental conflict). Please explain how both concepts differ.

Response 2: In lines 74-82, in defining co-parenting, it is explained how interparental conflict can affect and interfere with proper co-parenting, by undermining it.

We have tried to rewritten this part so it is better understood the difference between interparental conflict and co-parenting, and how these two factors may be interrelated, affecting one each other.

Comment 3: Much is presented on parental conflict, parenting skills, but less on the variables used in the analysis (specially on physical symptoms).

Response 3: We have tried to give more emphasis to physical symptomatology in order to try to balance the information about the main variables (lines 65-69).

Comment 4: I suggest being consistent with the concepts used. For example, it seems the authors use the concepts of parenting practices, parents’ socialization and parental functioning as interchangeable. This might be confusing for the readers.

Response 4: We have tried to be more consistent in the concepts used and to provide greater clarity. For this reason, we have rewritten the part where we talked about parenting practices (lines 74-82), so that it is better understood what we meant by the impact of co-parenting on parenting skills, which include emotion socialization. Regarding the concept of parental functioning, we understood that it could be confusing (line 124), so we have replaced it with emotion socialization.

Comment 5: In addition, in the introduction the author include statement such as “Between 5 - 25% of divorces occur with relatively high levels of conflict both during 46 and after the breakup (Hald et al., 2019)- line 46-7 ” or “…they occupy 90% of the resources of the family courts 53 (Hald et al., 2019; Ordway et al., 2020) and are often very challenging for the public 54 administration to handle (Haddad et al., 2016; Nikupeteri & Laitinen, 2022). Line 53-55 but they don´t clarify where these data come from. Are the from Spain (where the study was conducted)? If not, are they comparable to Spain?

Response 5: We understand the importance to consider the data at the national level. However, we had considered it at the European level, as we do not have information on high-conflict divorce rates in Spain. There is data on the most common type of procedure (mutual agreement or contentious) in 2021, but this does not necessarily imply a high level of destructive conflict. To make it clearer for the reader, though, we have specified that this is data from Europe.

Comment 6: Co-parenting is an important variable in the study; however, it is poorly defined. The central aspect of coordination between parents is missing. Moreover, it is surprising that the authors don’t mention any of the work by Feinberg or McHale

Response 6: We very much appreciate the recommendation on both authors, we have reviewed what they mention on co-parenting and tried to complete the information on co-parenting by also adding the aspect of coordination between parents (lines 74-76).

Hypothesis

Comment 7: About the hypothesis I suggest being more specific, for example including possible relationships between variables in their directions.

Response 7: We have tried to further clarify the relationships between the variables in the hypotheses, as suggested (lines 196-205).

Materials and Methods

Comment 8: The authors describe the sample as experiencing high levels of interpersonal conflict. Please explain how this variable was operationalized and assessed. My understanding is that this was a variable not assessed by the authors, but by the visiting program. It would be interesting to see how the distribution of levels of conflict in these families. In the discussion the authors say that the families in this sample present high levels of conflict. What does it mean?

Response 8: In lines 212-217 we explain that the sample comes from Family Visitation Centers, which are centers to which families come by judicial referral, in this case with high-conflict divorces. Therefore, this means that they are families with a high level of destructive conflict. The assessment for this was attending these centers, as specified in line 217.

Comment 9: For the instruments, please add reliability for fathers and mothers, and add information whether the instruments have been validated in Spain.

Response 9: We have added the information regarding the reliability for mothers and fathers in the instruments section of the method part. With regard to the Spanish version of the instruments, this was already indicated in those that have it, and was the one used in our study.

Results

Comment 10: Line 379: the authors report gender differences, but don’t specify whether they refer to male or female.

Response 10: It has been specified that the results refer to men (line 407).

Comment 11: Please include in the result section the information presented on line 467-8 (that coparenting score did not vary significantly among profiles). Please do the same for scores on resilience. The information that participants scored high in resilience in only provided at the end of the manuscript (in the discussion section).

Response 11: Data on co-parenting were not included, as they are contained in both Table 3 and Figure 1. We did not have much space due to the length criteria, so we chose not to repeat the information here, but to explain it better in the discussion.

Regarding resilience, this information is included in the results section (lines 432-440).

Discussion

Comment 12: The discussion of the results of profile 1 and use of coaching and rejection is not clear. The author say that these parents are more conscious of their own emotions. What evidence do they use to support this claim?

Response 12: In the first part of this section we said that parents with less symptomatology are more aware of their own emotions (Seddom et al., 2020) (lines 468-470). Therefore, in this case, Profile 1 parents, who have fewer symptoms, are likely to be more aware of their emotions.

Comment 13: On line 456-457 the authors say that parents in profile 3 have higher levels of conflict. Higher compared to what?

Response 13: We have modified this statement (line 484), as they are parents with more symptomatology, but we do not know if they have a higher level of conflict than the other two profiles.

Comment 14: Mikolajczak and Roskam refer to parental burnout, which is a specific concept. The way the authors (line 461-3) use this concept (naming it emotional burnout) and relate it to their findings is, in my opinion, not accurate.

Response 14: This section has been modified to make the explanation more accurate to our findings and the previous statements on the impact of parental symptomatology on parents’ emotion socialization patterns.

Comment 15: Please include more discussion on the profiles. Profile 3 is interesting, as it presents high symptoms but also high co-parenting.

Response 15: In this case, the differences in co-parenting between the three profiles are minimal, so the discussion have focused more on parental symptomatology. We have included a brief explanation of the co-parenting score in Profile 3 (lines 492-495), but, considering the results, it seems that symptomatology is more importan, as it makes the real difference in the distribution of the profiles and its impact on parents’ emotion socialization.

Other minor issues

Comment 16: Please revise sentence in line 60. What do the authors mean by “sensitivity”? In parenting research being sensitive is generally characterize as a positive aspect. Please revise table 5b. the title for the first column seems to be incorrect.

Response 16: We have changed the word “sensitivity” in line 61 because, in fact, what we really wanted to state was the lack of sensitivity that characterizes parents in high-conflict divorces.

As for Table 5b, we have rewritten the terms used in the first column.

Reviewer 2 Report

Comments and Suggestions for Authors

Author Response

We appreciate the thorough feedback given by the reviewers. It helped us improving both the content and writing of the manuscript. We would also like to acknowledge the positive words regarding the interest and implications of the study for future research and clinical practice.

All the adjustments in response to the suggestions of the reviewers are highlighted in yellow in the revised manuscript (complete manuscript resubmission) and this document indicates exactly on which lines they can be found.

Comment 1: The term "high-conflict divorces" is mentioned in the paper. It is recommended to provide a clearer definition of what constitutes a "high-conflict divorce" and how it differs from typical divorces in the introduction section. This will assist readers in better understanding the scope and context of the study.

Response 1: In the introduction (lines 47-49) we define what characterizes high-conflict divorces. We felt it was not necessary to repeat the characteristics in the method section, as we did not have much space due to the length limitations. However, in lines 212-213 and 216-217 it is explained that the high level of conflict is assessed on the basis of these parents attending the Family Visitation Centres.

Comment 2: Given that multiple scales (PSI-18, HADS, CAD-S, and ERPS) were used for data collection in the study, it is suggested to include tests for common method bias. This will help ensure the reliability of the research findings and bolster the persuasiveness of the conclusions.

Response 2: We have applied this test in order to increase the reliability of the study results (lines 355-359).

Comment 3: Latent Profile Analysis (LPA) has been employed as the primary data analysis method in the paper. It is recommended to elaborate on the reasons for selecting this method in the methodology section, including how it caters to the unique needs of this study and its advantages.

Response 3: The rationale for choosing this methodology and the advantages for this study are reflected in lines 180-184. We have not included this information again in the method section so as not to be repetitive, and also due to limitations on the length of the article.

Comment 4: The results section should be enriched with more specifics, such as the distinct manifestations and differences in parental emotional socialization patterns, along with potential explanations behind these differences. This will aid readers in gaining a deeper understanding of the findings and provide clearer guidance for future research.

Response 4: From line 458 (discussion) the differences in the profiles are explained in terms of symptomatology above all, although co-parenting is also mentioned, and we have tried to clarify the explanations for these differences regarding previous literature, and how they are related to parents’ emotion socialization. Due to length limitations, we considered not repeating this information in the results section.

Reviewer 3 Report

Comments and Suggestions for Authors

The topic of divorce is certainly rich in explanatory contributions in the literature, but normally it deals with the effects on the children of this transition, as the authors say.

This article instead focuses on parents who separate: how their physical and mental health affects their co-parenting and therefore the development of their children. This underlines its originality compared to most literature studies.

The study is supported by a rich bibliography - relevant and updated -.

More than half of the research sample is characterized by a low level of education and this could raise problems regarding the generalization of the results.

The authors do not mention this as it may represent a limitation of the study.

The sample problem is difficult to solve: this is why the authors are asked to write it clearly in the discussion and conclusions.

I would recommend the authors to use the spillover concept (Kats, Gottman, 1996), widely used in literature as it describes the object of the study well.

Author Response

We appreciate the thorough feedback given by the reviewers. It helped us improving both the content and writing of the manuscript. We would also like to acknowledge the positive words regarding the interest and implications of the study for future research and clinical practice.

All the adjustments in response to the suggestions of the reviewers are highlighted in yellow in the revised manuscript (complete manuscript resubmission) and this document indicates exactly on which lines they can be found.

Comment 1: More than half of the research sample is characterized by a low level of education and this could raise problems regarding the generalization of the results. The authors do not mention this as it may represent a limitation of the study.

The sample problem is difficult to solve: this is why the authors are asked to write it clearly in the discussion and conclusions.

Response 1: We have tried to include this information in the discussion (lines 529-534).

Comment 2: I would recommend the authors to use the spillover concept (Kats, Gottman, 1996), widely used in literature as it describes the object of the study well.

Response 2: We have included this concept in the introduction along with the Family Systems Theory (lines 84-92).